# Link between Viral Infections, Immune System, Inflammation and Diet

**DOI:** 10.3390/ijerph18052455

**Published:** 2021-03-02

**Authors:** Carlotta Suardi, Emanuela Cazzaniga, Stephanie Graci, Dario Dongo, Paola Palestini

**Affiliations:** 1School of Medicine and Surgery, University of Milano-Bicocca, EC, via Cadore, 48, 20900 Monza, Italy; carlottasuardi.bionutri@gmail.com (C.S.); stephanie_graci@outlook.it (S.G.); dario.dongo@me.com (D.D.); paola.palestini@unimib.it (P.P.); 2Bicocca Center of Science and Technology for FOOD, University of Milano-Bicocca, Piazza della Scienza, 2, 20126 Milano, Italy

**Keywords:** lifestyle, infection, immunity, microbiota, vitamin, dietary guidelines

## Abstract

The strong spread of COVID-19 and the significant number of deaths associated with it could be related to improper lifestyles, which lead to a low-grade inflammation (LGI) that not only increases the risk of chronic diseases, but also the risk of facing complications relating to infections and a greater susceptibility to infections themselves. Recently, scientific research has widely demonstrated that the microbiota plays a fundamental role in modulating metabolic responses in the immune system. There is, in fact, a two-way interaction between lifestyle, infection, and immunity. The immune response is compromised if nutrition is unbalanced or insufficient, because diet affects the intestinal flora predisposing people to infections and, at the same time, the nutritional state can be aggravated by the immune response itself to the infection. We evaluate the link between balanced diet, the efficiency of the immune system, and microbiota with the aim of providing some practical advice for individuals, with special attention to the elderly. A correct lifestyle that follows the Mediterranean model, which is especially rich in plant-based foods along with the use of extra-virgin olive oil, are the basis of preventing LGI and other chronic pathologies, directly influencing the intestinal microbiota and consequently the immune response.

## 1. Lifestyle Could Influence the Risk of Infections

Worldwide obesity has nearly tripled since 1975. In 2016, more than 1.9 billion adults, 18 years and older, were overweight and of these over 650 million were obese (39% overweight, and 13% obese) [1]. Excess weight and the abuse of food of animal origin are linked to the onset of chronic degenerative diseases affecting humanity [2,3].

The intermediary between diet and disease is low-grade inflammation (LGI), mediated by numerous substances that the body produces, such as inflammatory cytokines like Interleukin-6 (IL6), and also exogenous factors (such as smoking and pollution) that lead to the slow but inexorable erosion of DNA, cells, and their components (proteins and membrane lipids), causing diseases and premature aging [4].

The current life expectancy has increased, but LGI reduces not only the length of life but the health expectation. The term “inflammaging” refers to the fact that inflammation contributes to the aging process, increasing the probability that damage will occur [5]. LGI lays the foundation for many non-communicable diseases, such as heart attack, stroke, type 2 diabetes, arthritis, osteoporosis, depression, dementia, and tumors. The causes of LGI are largely attributed to our actions and lifestyle [5].

An unbalanced diet rich in calories, which leads to overweight and obesity includes excess of animal products, salt, saturated fats, and alcohol, and is poor in protective foods such as fruits and vegetables, legumes, whole grains, and fish. Furthermore, physical inactivity, poor quality of night-time sleep, stress, along with cigarette smoke, and exposure to exogenous pollutants contribute to a negative outcome [6].

LGI was also associated with an increased risk of subsequent infection, as demonstrated by Kaspersen [4] in a large cohort of healthy individuals.

In the case of viral infection like Covid-19, the aggravation is caused by different factors. The infection progresses for a few days with fever and dry cough (these are the most common symptoms), but in many cases the symptoms can be very mild or absent. A number of subjects have dyspnea (difficulty breathing), which can worsen in severe pneumonia, ARDS (acute respiratory distress syndrome), multiple organ dysfunction (MOD), sepsis, and septic shock, leading to death [7].

In healthy subjects, the virus multiplies in the body within the first week. Then, the immune system is mobilized with the production of inflammatory cytokines such as IL6, the same molecules that cause LGI. In the subjects that are already predisposed to a compromised immune system, it can lead to an uncontrolled inflammatory response, destroying the lung tissue, and circulating throughout the body causing damage to other organs. In extreme cases it can lead to death [8].

The subjects most at risk of experiencing the inflammatory storm are: (a) people who are already predisposed to LGI for various reasons such as hypertension, cardiovascular diseases, tumors, diabetes, and obesity [9]. In smokers and those who abuse alcohol, inflammation is already present, which adds to the cytokine storm thereby aggravating the prognosis [10]; (b) the elderly, who, for physiological reasons, have a slightly higher state of inflammation, which takes very little to create an imbalance [11]; (c) subjects genetically predisposed to producing higher cytokine levels. This would also explain why the disease affects, to a small extent, young people, who may not show visible signs even if they are overweight [12].

It is known that an unbalanced lifestyle leads to low-grade inflammation, which not only increases the risk of chronic diseases but also the risk of experiencing complications related to this infection (and probably also a greater susceptibility to infections themselves) [6,13].

There is a bidirectional interaction between nutrition, infection, and immunity: the immune response is compromised if nutrition is insufficient, thereby predisposing people to infections.

Moreover, a poor nutritional status can aggravate the infection’s immune response due to micronutrient deficiencies and suboptimal assumptions. It is known that malnutrition induces a reduction in immune cells number, in particular T- and B-cells, leading to leucopenia. This decreased number of immune cells contributes to the impairment of the immune response in malnutrition [14,15].

On the other hand, non-communicable diseases have been correlated with poor nutrition and immunodeficiency. Indeed, inadequate glycemic control is associated with several infections, such as respiratory infections. Moreover, in obese individuals, two factors are known that make them more susceptible to viable infections; LGI due to excess of white adipose tissue, and increased epithelial permeability, which could permit rapid virus shedding from the tissue and, consequently, faster spreading [15].

## 2. Micronutrients’ Roles in the Immune System

Micronutrient deficiencies and suboptimal intakes may be more likely to be detected in children and the elderly. This can affect the risk and severity of infections. The nutritional status of an individual can predict the clinical course and outcome of some infections, such as pneumonia. Resistance to infections can be improved by adding the deficient nutrients within an optimal diet, resulting in restored immune functions. Micronutrients play a vital role in the whole immune system, but those most important to support immunocompetence are vitamins A, C, D, E, B6 and B12, folic acid, and iron, copper, selenium and zinc [16]. 

Vitamin D—Vitamin D hypovitaminosis is a widespread disorder. Clinical manifestations of vitamin D hypovitaminosis include musculoskeletal and nonmusculoskeletal disorders, an increased risk of respiratory infections, diabetes mellitus, and cardiovascular diseases. The prevalence of vitamin D hypovitaminosis varies widely between countries (from 30% to 90%), based on the threshold value used within the specific regions [17,18]. The main source of vitamin D for humans is sunlight. Any factor that decreases the transmission of UVB radiation on the Earth’s surface or anything that interferes with the penetration of UVB radiation into the skin will affect the skin’s synthesis of vitamin D3 [18]. Few specific foods naturally contain vitamin D (from the richest food): cod liver oil; oily fish, particularly mackerel, herring, tuna, and salmon; egg yolk; mushrooms (which is the only vegetable source of vitamin D); fatty cheeses, and butter [19,20]. Through several mechanisms, vitamin D can reduce the risk of infections. 

Active vitamin D (1,25(OH)2 D), a steroid hormone, has profound effects on human immunity. Liu et al. [21] by studying tuberculosis, suggested a possible mechanism; probably when a macrophage is infected with tuberculosis it stimulates the cell to increase the production of 1,25-dihydroxyvitamin D3 (1,25 (OH)2 D3) and increases the expression of the vitamin D receptor. In combination, the gene expression of the bactericidal protein cathelicidin and defensins can lower the viral replication rate [18,22]. Another supposed mechanism involves the effect of vitamin D on reducing pro-inflammatory cytokine concentrations, which produce inflammation that can damage the lining of the lungs, causing pneumonia as well as increasing anti-inflammatory cytokine concentrations [22].

Data support a relationship between the toll-like receptor (TLR) and innate immunity mediated by vitamin D, suggesting that differences in the ability of human populations to produce vitamin D may contribute to the different susceptibility to microbial infections [21]. In addition, 1,25 (OH)2 D acts as an immune system modulator, preventing excessive expression of inflammatory cytokines and increasing the “oxidative burst” potential of macrophages. More importantly, it significantly stimulates the expression of potent anti-microbial peptides, in neutrophils, monocytes, natural killer cells, and epithelial cells that line the respiratory tract, where they play an important role in protecting the lung from infections [23]. It should be noted that the inadequate intake of vitamin D along with food is endemic in the elderly during winter [24,25,26]. The elderly produce about 25% of vitamin D compared to young individuals after being exposed to the same amount of sunlight [27]. For these reasons, vitamin D deficiency could predispose these patients to respiratory infections [28,29]. Some retrospective studies demonstrated a correlation between vitamin D status and COVID-19 severity and mortality, while other studies did not find the correlation when confounding variables are adjusted. There is not enough evidence on the association between vitamin D levels and COVID-19 severity and mortality. Therefore, randomized control trials and cohort studies are necessary to test this hypothesis [22,30,31,32].

Vitamin C—This is an effective antioxidant against ROS (reactive oxygen species), found when pathogens are killed by immune cells. It regenerates other important antioxidants like glutathione and vitamin E, which promote the synthesis of collagen thus supporting the integrity of epithelial barriers. It also stimulates the production, function, and movement of leukocytes (e.g., neutrophils, lymphocytes, phagocytes) and has a role in antimicrobial activity and chemotaxis [16,33,34]. It is known that supplementing with this vitamin could reduce the duration and severity of upper respiratory infections (most of which are assumed to be due to viral infections) [29,35]. Although evidence is currently weak in relation to the utility of vitamin C against COVID-19 infections, low vitamin C status has been discussed as an adjuvant measure to aid in individuals with the common cold and also pneumonia, and positive effects were found in some intervention trials, such as shortening the duration and severity of colds [36,37]. Due to a lack of evidence against COVID-19, there are limited recommendations for vitamin C intake [37].

The foods richest in Vitamin C include peppers, blackcurrant, parsley, cruciferous vegetables (cabbage, savoy cabbage, and broccoli), kiwi, and citrus fruits [19,20].

Vitamin A—This helps maintain the structural and functional integrity of the cells of the mucosa (e.g., skin, respiratory tract, etc.). This vitamin is important for the normal function of innate immune cells (macrophages, neutrophils). It is also necessary for the proper functioning of T and B lymphocytes, therefore making it necessary for the generation of antibody responses to the antigen. It is involved in the development and differentiation of Th1 and Th2 cells and supports the anti-inflammatory Th2 response [16,33,36].

In general, studies investigating the efficacy of vitamin A supplementation on improvement of immune responses to vaccines have produced conflicting results. However, it is suspected that pre-existing vitamin A stores play crucial roles in resulting responses [36]. In light of its pulmonary and immunological roles, oral supplementation of vitamin A is currently being investigated in the treatment of COVID-19 alongside a host of other antioxidants [32].

The foods richest in vitamin A include cod liver oil, liver, dried apricots, carrots, hot peppers, and pumpkin [19,20].

Vitamin E—This is an important fat-soluble antioxidant: it protects the integrity of cell membranes from damage caused by ROS and improves IL-2 production, T-mediated cell functions, and lymphocyte proliferation; it also optimizes and improves Th1 and suppresses Th2 response [34,38].

The role of vitamin E in the prevention of infections such as influenza has been discussed, but well-controlled studies in humans are lacking. Supplementation in humans with vitamin E seems to restore IL-2 production, improving T-cell proliferation and immune system functioning [36]. Indeed, it has been suggested that a combination of vitamins C and E may be a useful antioxidant therapy for cardiac complications of COVID-19. However, there is little evidence to date on the utility of vitamin E as a prophylactic or therapeutic agent against COVID-19 [37] and, as the pandemic evolves further, research may unravel the potential benefits. The foods richest in vitamin E include vegetable oils (wheat germ oil, sunflower, corn, and extra-virgin olive oil), basil, hazelnuts, and avocado [19,20].

Vitamins B6 and B12—These vitamins help regulate inflammation. They play an important role in the production of antibodies, cytokines, and in the proliferation and differentiation of lymphocytes. They maintain Th1 immune response [16,34,38].

Vitamin B6 can be found in tempeh, whole grains, shiitake mushroom, and liver. Vitamin B12 can be found in liver, aged cheeses, seafood and fish (tuna, cod, sardines and mackerel), egg yolk, and milk [20]. 

Folic acid—Maintains innate immunity. It is important for the antibody response to antigens. It supports Th1-mediated immune response [38,39].

This vitamin can be found in liver, asparagus, broccoli, Brussel sprouts, artichokes, cauliflower, whole grains, and legumes [19,20].

Trials with a larger number of participants and stronger epidemiological designs are needed to provide more definitive information about the roles of B vitamins in the immune system and infection [36].

Metals (iron, zinc, selenium, copper)—are important in modulating the immune responses (in the differentiation and proliferation of T lymphocytes, in the production of antibodies and in cellular immunity) with an antioxidant action (selenium and zinc) [16,29,34,36,38,40]. Regarding the iron, a significant difference in mean ferritin levels was found between survivor and non-survivor COVID-19 patients, but not in hemoglobin levels. Future studies should explore the impact of iron metabolism and anemia in the pathophysiology, prognosis, and treatment of COVID-19 [41]. In the elderly, low zinc status doubles the mortality rate, due to pneumonia [36].

No effects of selenium supplementation were observed on overall survival, the duration of mechanical ventilation, or ICU stay, which also raises caution in interpolating findings from sub-clinical markers to harder endpoints, and further studies are needed [36]. At the last, more research is required to determine whether higher intakes of dietary copper may benefit immune functions against viral infections [37].

Selenium is found in tuna, octopus, and sea bream. Iron is found in liver, beef, horse meat, eggs, some fish (anchovy, mullet, sardine, tuna), legumes and nuts, herbs, peppers, and mushroom. Zinc is found in fish, cereals (wheat germ and oats), mushroom, legumes, nuts, and seeds. Copper can be found in liver, oysters, cashews, and walnuts [20].

In addition, two non-essential amino acids have been shown to be relevant to the immune system:(1)Glutamine is an amino acid important to the immune cells such as lymphocytes, natural killer cells, and the proliferation of macrophages. Glutamine is important for the synthesis of glutathione [42,43]. Foods that are particularly rich in amino acid are eggs, beef, milk, tofu, and white rice [21].(2)Arginine improves the function of T lymphocytes. It is the precursor of nitric oxide which has an important role in coagulation, vasodilation, vascular permeability and the destruction of microbial pathogens [43,44]. Particularly rich foods are pine nuts, peanuts, eggs, and white meat [20].

Fatty acids ω-3 (EPA and DHA)—These are important precursors of molecules designed to promote the resolution of inflammation. They also improve bacterial killing by macrophages and increase the regeneration of fabrics [43,44]. While it is known that ω-3 fatty acids can beneficially interact with the COX enzymes, it also is not clear if fish or fish oil consumption may be beneficial against SARS-CoV-2 infection [37].

EPA and DHA are found in cod liver oil, turkey, and in some fish (salmon, cod, tuna, and sardines) [20].

## 3. Prebiotics, Probiotics, and the Immune System

The intestinal microbiota consists of billions of microorganisms (bacteria, viruses, fungi, protozoa) from different species that live in symbiosis with our organism. These microorganisms play a fundamental role in health, acting as an intermediary between the external and internal environments [45].

The intestine of a healthy individual is populated by billions of bacteria belonging to over 800 different species, mostly called commensals or “good bacteria” (lactobacilli and bifidobacteria in particular). These, as a whole, make up the bacterial microbiota, interacting not only with each other but also with other microorganisms (fungi, archea, etc.) as well as with the host [46].

In recent years, scientific research has widely demonstrated that the microbial community present in the intestinal tract plays a fundamental role in modulating metabolic responses in the immune system. Recent data has shown that there is a relationship between diet, immune system, and intestinal microbiome [47]. It is known that diet not only influences the composition of the intestinal microbiota but also that the microbiota and its products have an effect on the host [48,49].

Many bio-functional activities depend on the health of the microbiome. The microbiota modulates the metabolism of some nutrients and the production of some neurotransmitters (serotonin, etc.) reducing the possibility of developing allergies and/or intolerances. In addition, it produces a significant amount of vitamins and important short chain fatty acids (SCFA, e.g., butyric and propionic acid originating from the microbial fermentation of fiber) involved in energy homeostasis and therefore in the modulation of body weight (loss or gain), in disorders associated with obesity and overweight and in the suppression of inflammatory signals [46,50]). 

An important advance in the fields of immunology and gut microbiology has emerged in the last decade, as findings from many laboratories have shown that the microbiota influences the activity and expression of agents involved in the immune response (cytokines, lymphocytes, dendritic cells, pro/anti-inflammatory agents etc.) [49].

Scientific evidence accumulated in recent years suggests that the microbiota is an active leader of the bidirectional modulation of the intestine–brain axis in which an unsettled intestine sends signals to the brain, and vice versa. Gastrointestinal distress can be caused by anxious states, stress, or depressive states [51]. 

Finally, the microbiota is involved in the health and correct functioning of various organs and systems [52]. By studying the link between diet and microbiome, currently we know that some components of food, such as fibers, which are not digested and absorbed by the host, act as nutrients of the microbiota, which by metabolizing them produces short chain fatty acids (SCFA). SCFA, through intestinal receptors, send “signals” to the central nervous system with the aim of modulating the energy homeostasis, the metabolism of carbohydrates and lipids with the suppression of inflammatory signals [46,52]. Previous studies have shown that a diet rich in fiber induces a certain composition of the microbiota with an increase in the production of SCFA, compared to a diet low in fiber. Dietary changes towards a western diet rich in fats and proteins and low in fiber, fruits, and vegetables, increase the risk of chronic immune diseases, such as inflammatory bowel disease [53].

Aside from the indirect effects of SCFA production by the microbiota, some fibers have been reported to have an immunostimulatory effect on immune cells. An example is the β-1,3-glycan that is present in mushrooms along the pectin’s arabine, present in the peel of apples, pears, apricots, and plums [54].

Another important link between man, diet, and microbiota is the aromatic hydrocarbon receptor (AhR) and immune system axis. Diet can influence this axis by providing sources of ligands for AhR. AhRs are important anti-inflammatory receptors expressed by a large number of immune cells and the signaling mediated by these receptors could integrate the effects of the environment (diet) along with microbial metabolism and immune response [55].

The main exogenous ligands for AhR derive from edible plant tissues, e.g., vegetables, fruit, tea, and herbs. Several studies have shown that especially polyphenols can create an activating effect. Polyphenols consists of a family of about 5000 natural organic molecules present in plants best known and studied for their beneficial properties on human health, such as quercitin, resveratrol, epigallocatechin, anthocyanins, and tyrisol [55,56]. Depending on their size and structure, polyphenols can be absorbed in the small intestine, but the bioavailable amount is generally small (about 5–10%) while the highest percentage reaches the colon unchanged where it is metabolized by the intestinal microbiota. These are modified and non-modified molecules that act as AhR ligands [57,58]. Another example of microbial metabolites that can activate AhR are tryptophan’s metabolites. These are essential amino acids that are absorbed (70–95%) by our intestine while what remains is metabolized by the microbiota, creating indole derivatives, which are strong activators of AhR [59].

Maintaining a healthy intestinal microbiota is therefore very important. If the balance of the intestinal microflora is disturbed due to the administration of antibiotics, an unbalanced diet, smoking, alcohol, environmental pollutants, excessive consumption of drugs, or laxatives, it can easily undergo the invasion of pathogens or viruses predisposing our body to a greater risk of infections. This increases intestinal permeability, which causes a weakening of the immune system [48,60]. In this case, we find ourselves in an increased state of inflammation and intestinal dysbiosis, a circumstance in which the bacteria normally present in the intestine are greatly reduced. In fact, a perturbation of gut microbiota integrity has been found in SARS-COVID patients [47].

For example, we may be deficient in *Clostridium* and *Eubacterium*, bacteria known to metabolize phenolic compounds into bioactive metabolites that are absorbed by the host, or in *Lactobacilli*, bacteria known to metabolize tryptophan-forming metabolites that can induce an immune response though AhR [61].

It is clear that maintaining the correct microbiota is closely related to optimal well-being and therefore to an efficient immune system. Consequently, the strategies are:(a)A correct diet rich in prebiotics–polysaccharides (non-digestible fruit- and galacto-oligosaccharides); a specific subgroup of MACs (carbohydrates accessible to microbiota) of different lengths, which modify the composition of the intestinal microbiota by selectively promoting the growth of *Bifidobacterium* and *Lactobacillus* [62,63]. MACs are found in fruits, vegetables, whole grains, and legumes (key foods of our Mediterranean diet as described in the model of healthy diets from sustainable food systems proposed in The Lancet [64]. Currently, the term prebiotic also refers to polyunsaturated fatty acids such as linoleic acids, phytocompounds, and phenolic compounds that can also positively modulate the intestinal microbiota.(b)Adequate physical activity and adequate rest.(c)Possible integration of probiotics: live microorganisms which, when administered in adequate quantities and times (at least one billion a day for a variable treatment ranging from 3–4 weeks to about 3 months), can exercise beneficial functions for the organism [65]. The main probiotics are found in specific supplements or in fermented foods that we find all over the world as an ancient method of preserving food. Some examples are kefir fermented milk, Korean kimchi, and Japanese miso.

The study of probiotic products is constantly evolving. To the present knowledge, the mechanisms of specific strains on the prevention and eventually the cure of pathologies are still not clear. Researchers are focusing on several formulations. A 2018 study [66] highlighted the anti-inflammatory effect of a formulation with *L*. *Rhamnosus*, *B. lactis* and *B. longum.* In a very recent 2020 study, the authors report that the administration of *Bifidobacterium Longum* BB536 and *Lactobacillus rhamnosus* HN001 improves the symptoms and severity of IBS, restores intestinal permeability, and the microbiota [67]. For patients suffering from dermatosis, a 2010 McFarland meta-analysis highlights the effectiveness of the action of *Saccharomices Boulardii* in adults [68].

When a food or supplement consists of the association between a prebiotic and a probiotic, we get a product called a symbiotic. Scientific evidence supports the effectiveness of prebiotics thanks to a varied and balanced diet and of probiotics in improving our overall state of health. This not only acts on gastrointestinal diseases, but also by improving our immune defenses, strengthens the structure and function of the intestinal barrier, as well as the well-being of the intestinal microenvironment. Dosage, route of administration, inter-individual variability, and strain-specific properties are just some of the variables to be studied in depth before considering the results obtained from studies that are reliable and reproducible [69].

## 4. Focus on the Elderly

Human aging is a process that begins after reaching sexual maturity. This is determined by degenerative changes with various manifestations such as gradual organic dysfunction, loss of tissue function, an increase in the aged (senescent) cells population, and reduction of repair capacity, which collectively includes an increased risk of disease and death. In addition, a main feature of aging is the presence of a constant but low degree of inflammation called inflammaging [5].

Moreover, we must remember that between 4.9% and 27.3% of all seniors over the age of 65 all over the world are defined as frail (frailty phenotype). Frailty means lack of skills to cope with the daily stressors associated with aging [70]. 

The frailty phenotype of the elderly is based on the analysis of five criteria; exhaustion, low physical activity, weakness, weight loss, and sedentary behavior. The latter two factors are related to a low nutrient supply and are certainly the main factors responsible for fragility [71].

Several nutrients have demonstrated their role in physical maintenance in the elderly through the optimization of bone and muscle health and it is now known that nutritional deficiencies are constantly connected to a physical decline [70].

From all this, it is easy to conclude that the elderly are part of the population most at risk of contracting infections because aging leads to immuno-senescence and the lowering of immune defenses, which is enhanced if the elder is malnourished, hence frail. Therefore, optimal nutrition is the focal point in maintaining a better immune defense. 

The energy needs of an elderly person over 65 years old and without relevant pathologies are lower, around 1600–2100 Kcal/day in women and 1750–2300 Kcal/day in men, depending on physical activity. Inside these values, all macro- and micro-elements must be correctly distributed [72]. With respect to the need for vitamins and microelements, the guidelines indicate a greater need for vitamins K and B6, while paying particular attention to vitamins B12 and D for the geriatric age group. The elderly are the group most at risk for deficiency of the latter vitamin due to a lack of direct exposure to sunlight and decreased endogenous synthesis. Vitamin deficiencies in the elderly are often not clinically detectable, but have been frequently associated with disorders such as anorexia, impaired cognitive status, depressive syndromes, etc. [70,71].

Finally, it is crucial to underline that the elderly take one or more medications each day, and many of these can decrease the bioavailability of many vitamins and salts (for example antacids, colchicine, laxatives, levodopa, metformin, vitamin B12, broad-spectrum antibiotics, antiepileptics, laxatives, vitamin K, diuretics, vitamin B6, and salts) [72,73].

In this age group, the contributions of calcium, sodium, potassium, iron, and zinc deserve particular attention. 

The recommended calcium intake level is higher than expected for adults (1200 mg/day) as the loss is expected in the elderly due to bone demineralization, responsible for osteoporosis. [72,73].

In the geriatric age group, there is high prevalence of arterial hypertension and a higher risk of cardiovascular failure, as well as cerebrovascular risks. For these reasons, the sodium intake should be reduced (target 1.6 g/day) in this group while the potassium requirement does not differ.

The prevalence of anemia increases with age and represents an important health problem. The causes that contribute to the onset of iron deficiency are: reduced or inadequate intake within the diet, reduced absorption, nutritional deficiencies of vitamin B12 and folate, loss through occult bleeding, and the intake of some drugs. Therefore, in elderly people, the recommended iron intake values are 10 mg/day. Vitamin C promotes iron absorption, while vitamin B12 and folate play an important role in the prevention of anemia [72,73].

Zinc deficiency can be determined by a reduced intestinal absorption, reduced intake of animal proteins, as well as the increased loss due to current pathologies and/or pharmacological therapies [72,73]. Consequently, the vitamin and microelement requirements can be more satisfied if the diet of the elderly person is often varied. Such a diet consists of foods from both animal and vegetable origin, especially fresh and seasonal fruit and vegetables.

In addition to being rich in nutrients, the latter foods also contain molecules and phytocompounds that are not essential for human health but certainly have beneficial properties for our bodies [72,73].

## 5. Additional Advice for a Plant-Based Diet

The SINU (Italian Society of Human Nutrition) position paper on Vegetarian Diets of 2012 [72] and the American Dietetic Association [73] provide all the information to be taken into consideration for those following a vegetarian or vegan diet.

The analyzed data show that most vegetarian diets provide an adequate nutritional intake for all age groups. However, it is important to monitor the status of key nutrients [72,73] (proteins, vitamin B12, calcium, iron, zinc, vitamin D, and omega-3 fatty acids), which may not always be present in optimal quantities in some types of vegetarian diets.

Since the digestibility of vegetable proteins is lower than that of animal proteins, it may be appropriate for vegetarians and vegans to take a slightly higher amount of proteins (5–10%) than suggested for the general population.

Vegetarians and vegans should also supplement their diet with a reliable source of vitamin B12 (fortified foods or supplements). Some studies have shown that the absorption of vitamin B12 is often less than 50% [72]. European Food Safety Authority (EFSA) estimates an absorption of 40% and considers an adequate intake to be 4 μg/day or more [74].

As for iron, it is recommended that nonmeat-eaters take 80% more than the omnivorous (OMN): 10 mg for adult males (versus the 7 mg recommended for the OMN) and 18 mg for females (versus the recommended 10 mg) [75,76,77]. It is estimated that compared to an OMN diet with a theoretical bioavailability of iron equal to 18%, the bioavailability of iron in a vegetarian (VEG) or vegan (VG) diet equals 10%.

Vegetarians should be sure to adopt a diet that respects the recommended intakes of calcium foreseen by the LARN IV revision [72]. In particular, VEGs should pay particular attention to the intake of food products that are rich in calcium (low-oxalate and phytate vegetables, soy-based foods, fortified vegetable drinks, calcium-rich waters, some types of nuts, and oil seeds). Interesting studies indicate that calcium chloride and calcium sulphate, used to produce tofu, have a bioavailability comparable to that of cow’s milk. The bioavailability of calcium that is present in mineral waters is equal to or greater than that of milk [78,79].

Any integration of vitamin D must be carefully considered in all the cases where insufficient endogenous synthesis is suspected.

With a plant-based diet, it is important to ensure an adequate supply of essential omega-3 fatty acids thanks to the habitual consumption of good sources of linolenic acids (nuts, flax seeds, chia seeds, and oils derived from them) considering that an average portion of nuts and oil seeds is about 30 g [72].

For Italians, it is possible to follow a healthy and nutritionally adequate vegetarian diet by choosing from the wide range of plant-based foods characteristic of our tradition (cereals, legumes, vegetables, fruit, seeds, nuts, extra-virgin olive oil). The consumption of foods typical of other cultures (e.g., soy products) or processed foods (e.g., seitan, extruded soy) is a matter of personal choice and is not necessary for an adequate and balanced vegetarian diet.

## 6. Conclusions

There are numerous nutrients that support our immune system, therefore a healthy and balanced diet is essential to staying healthy and facing infections. From what has been said, the foods that we must give priority to are those of the Mediterranean diet. In particular, seasonal fruit and vegetables should never be missing from our tables. It is recommended to eat at least five portions a day in order to meet the minerals and vitamins (especially vitamin C and vitamin A) requirements useful in strengthening the immune system. For carbohydrate sources, which must cover about 45–60% of an adult’s daily caloric intake and should be taken mainly in the form of complex carbohydrates, we should consume cereals, preferably whole grains rich in vitamin B6, along with folic acid and selenium. For protein sources, which must cover about 10–15% of daily calories, give priority to legumes (2–4 times a week), which are rich in fiber and a source of important micronutrients for the immune system (including folic acid, iron, and copper). Research shows that there is an inverse correlation between the consumption of legumes and the risk of chronic degenerative diseases, such as cardiovascular diseases, diabetes, obesity, and metabolic syndrome [80]. Fish is a good source of omega 3 (ω-3, EPA and DHA), recommended to be consumed 2–3 times a week (fresh or frozen). Meat is a good source of folic acid, iron, zinc, and arginine that should only be consumed 1–2 times a week. White meat (chicken, turkey, or rabbit) is better for you than red meat (bovine, pig, sheep, horse, and wild game). Eating 2–4 eggs per week supplies us with essential vitamin B12, selenium, iron, glutamine, and arginine. Aged cheeses are a good source of numerous nutrients essential to the immune system, but they are also very fatty and therefore should be consumed with caution. The average portion recommended by the guidelines is 50 g, which must not exceed three times a week. This includes cheeses with a content of less than 25% fat the recommended portion size of which can accordingly be increased to 100 g. [81]. Select extra-virgin olive oil as a seasoning fat, with a recommended portion being a tablespoon (10 mL) per meal or a total of 20 mL per day for a 1500 Kcal/day diet increasing to 40 mL per day for a 2500 Kcal/day diet [81]. Extra-virgin olive oil is rich in monosaturated fatty acids important for the prevention of cardiovascular diseases. Nuts and oil seeds can also be added to a diet in small quantities. The recommended portion for nuts is 30 g per serving, which equates to 7–8 walnuts, 15–20 almonds, or 3 tablespoons of peanuts. The recommended portion for oil seeds (sunflower, pumpkin, flax, sesame seeds, etc.) is three tablespoons. These foods have a high caloric density so they should be consumed with caution. For a 2000 Kcal diet, the new guidelines recommend consuming no more than 30 g of nuts a day, not exceeding twice in a week. This could also be consumed in a combination of 10 g of nuts and oil seeds per day.

Select semi-skimmed milk over whole milk and in cases of lactose intolerance, choose vegetable drinks with no added sugar, fortified with calcium and vitamin D. The recommended portion size for yogurt or other types of fermented milk is 125 mL/gr, three times a day. In addition to food sources, alternative methods of consuming vitamin D, whenever possible, is exposing arms and legs to the sun for 15–30 min to promote endogenous synthesis.

In conclusion, now more than ever, broader access to healthy foods should be considered a priority and individuals should be aware that healthy eating habits can help reduce susceptibility and complications due to viral diseases.

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
