# Peer review of "Link between Viral Infections, Immune System, Inflammation and Diet"

_ijerph, 2021, doi:10.3390/ijerph18052455_

Round 1
Reviewer 1 Report
The review article by Suardi and colleagues deals with a very current topic such as the link among inflammation, immune response, microbiota and diet.
However, while the title anticipates a discussion of the role of this cross-talk in viral infections, there is nothing about the latter except a hint of SARS-CoV-2 infection in the first paragraph. Addressing various viral infections could have been the distinctive element of this review, considering that several recent review articles on covid-19 and nutrition have been published.
Moreover there is a lot of evidence on the association of some micronutrients, such as vitamin D, with Covid-19 that is not mentioned.
It sounds like a list of many, too many information not linked in an organic discourse.
The English language needs revision.
Reviewer 2 Report
REVIEW COMMENTS
The manuscript titled “Link between viral infections, immune system, inflammation, and diet” makes an analysis between these topics to provide some recommendations. I believe that in the actual context of COVID infection, this is a suitable manuscript. Moreover, the topic fits within the scope of IJERPH. Overall, the English style and grammar need to be checked, so I advise the authors to check this aspect carefully. Moreover, some text parts need to be written more as a review from a scientific journal than a personal opinion or a list from a textbook. Based on my evaluation, authors are required to fulfill these major Revisions to make this manuscript suitable for publication in IJERPH.
- Line 34: Please remove “our”.
- Line 41: Please correct: “The causes of LGI are largely attributed to (...)”. What do the authors mean by “our actions”? This is a highly subjective term, so please be more specific or find a suitable replacement.
- Lines 43-44: Can fruit and vegetables be considered as “factors”?
- Lines 44-45: I don’t think this sentence is correctly written. It seems that physical inactivity, poor quality of night sleep, stress, etc., are also protective “factors” at the same level as fruits and vegetables.
- What is the connection between the paragraph in lines 42-46 and paragraph in lines 47-52? I think the authors should find a nexus between these ideas.
- Line 52: Please correct: “sepsis, and septic shock, leading to death”.
- Line 54: Please correct: “In healthy subjects, the virus multiplies in the body within the first week. Then, the immune system (...)”
- Lines 56-58: Please correct: “(...) response, destroying the lung tissue, and circulating throughout the body (...)”
- I feel the section “Lifestyle could influence the risk of infections” is not associated with the title of the text nor enough information, as an introductory section, is presented to overall indicate why lifestyle could influence the risk of infections. For instance, authors could add more information in lines 74-76 about why “poor nutritional status” aggravates the immune response. Authors should be more specific, indicating what is a “poor nutritional status”.
- Lines 78-85 could be better placed at the beginning of the 2nd
- I believe the manuscript would be better if the authors could find relevant information regarding the relationship between COVID-19 and each of the highlighted proteins, as COVID is still an emerging and highly attractive topic.
- Line 185: Please find a suitable replacement for “flora” as this term is updated from what is now called “microbiota”.
- Lines 200-201: SCFAs don’t originate in the fiber, they originate from the microbial fermentation of fiber.
- Lines 194-207 need to be rewritten more sequentially. Avoid the use of semicolons.
- Line 251: Please write scientific names using italics.
- Line 263: Please correct: “The Lancet”.
- Lines 276-277: Please write scientific names in italics.
- Line 312: Please replace “Let’s analyze” with a suitable expression since this term is not very scientifically sound.
- Lines 327-346 needs to be rewritten, not as a list, but as a concise text.
Round 2
Reviewer 1 Report
The manuscript has been improved.
Reviewer 2 Report
I appreciate the time taken by the authors to improve their manuscript. I believe it is now suitable for publication in IJERPH.